# NaviCache: Test-Time Self-Calibration Caching for Video Generation

**Zheqi Lv** [1 2]  **Zhibo Zhu** [1]  **Jinke Wang** [1]  **Qi Tian** [3]  **Shengyu Zhang** [1]  **Zhengyu Chen** [4]  **Chengxi Zang** [2]
**Zhou Zhao** [1]  **Fei Wu** [1]

https://github.com/HelloZicky/NaviCache

## Abstract

Video Diffusion Models (VDMs) is constrained by immense computational costs. While offline calibration-based acceleration suffers from calibration data dependency, prohibitive calibration duration, and susceptibility to distribution shifts, offline calibration-free methods eliminate these hurdles. However, since they rely on instantaneous zero-order approximations where the mapping between input and output differences varies in real-time, they are susceptible to observational noise and ignore the intrinsic momentum within the diffusion trajectory. In this paper, we propose **NaviCache**, a plug-and-play test-time self-calibration method re-conceptualizing feature evolution as an Inertial Navigation System (INS) problem. NaviCache bridges the fundamental domain gap and the non-stationary nature of diffusion by modeling the relative coupling between input and output variations. We introduce a dual-state estimation architecture that adaptively tracks the feature change ratio and its latent drift, initialized via a specialized Initial Alignment phase. By integrating a time-dependent noise schedule with an uncertainty-aware Measurement Update mechanism, NaviCache provides a theoretically grounded mechanism for error-bounded computation skipping. Extensive experiments on the HunyuanVideo, Wan, and Open-Sora series demonstrate that NaviCache exhibits more accurate error judgment for computation skipping and achieves outstanding comprehensive performance.

[1]Zhejiang University, Hangzhou, China [2]Cornell University, New York, USA [3]Tencent Hunyuan, Shenzhen, China [4]MeiTuan LongCat, Beijing, China. Correspondence to: Shengyu Zhang <sy_zhang@zju.edu.cn>.

*Proceedings of the $43^{rd}$ International Conference on Machine Learning*, Seoul, South Korea. PMLR 306, 2026. Copyright 2026 by the author(s).

## 1. Introduction

With the advent of video diffusion models (VDM) such as HunyuanVideo (Kong et al., 2024), Wan (Wan Team et al., 2025), and Open-Sora (Zheng et al., 2024; 2025), the field of video generation has witnessed rapid development. Despite this success, the iterative sampling process remains a computational tax requiring dozens of forward passes through massive architectures that hinder real-time deployment. Consequently, recent research on video generation acceleration has primarily bifurcated into: (i) Offline calibration-dependent methods. This category includes training-based methods or those relying on statistical priors obtained from curated datasets. For instance, *TeaCache* (Liu et al., 2025) uses polynomial fitting of residuals, while *MagCache* (Ma et al., 2026) utilizes a unified magnitude law. However, these methods necessitate a calibration cost and susceptible to distribution shifts between the calibration set and actual deployment scenarios. Furthermore, they are prone to multi-value fitting issues—where a single $x$ maps to multiple $y$ values—as depicted by the "raw points" in Figure 1. (ii) Offline calibration-free methods. To address these limitations, offline calibration-free approaches like *EasyCache* (Zhou et al., 2025) have emerged. They eliminate the need for calibration datasets, avoiding performance degradation caused by distribution shifts and saving preprocessing time. However, existing offline calibration-free methods use zero-order approximation at test time, which reduces performance due to the delayed update.

In this work, we focus on offline calibration-free methods and offer a transformative perspective: visualizing the relationship between input variations and output responses reveals a manifold resembling a kinematic navigation track. As visualized in Figure 1 (Raw Points), the relationship between input variations and output responses reveals a manifold resembling a kinematic navigation track, suggesting that feature evolution possesses underlying momentum rather than random jitters. This suggests feature evolution is a structured trajectory with underlying momentum rather than random jitters. Nevertheless, bridging control and navigation theory with video generation poses two significant

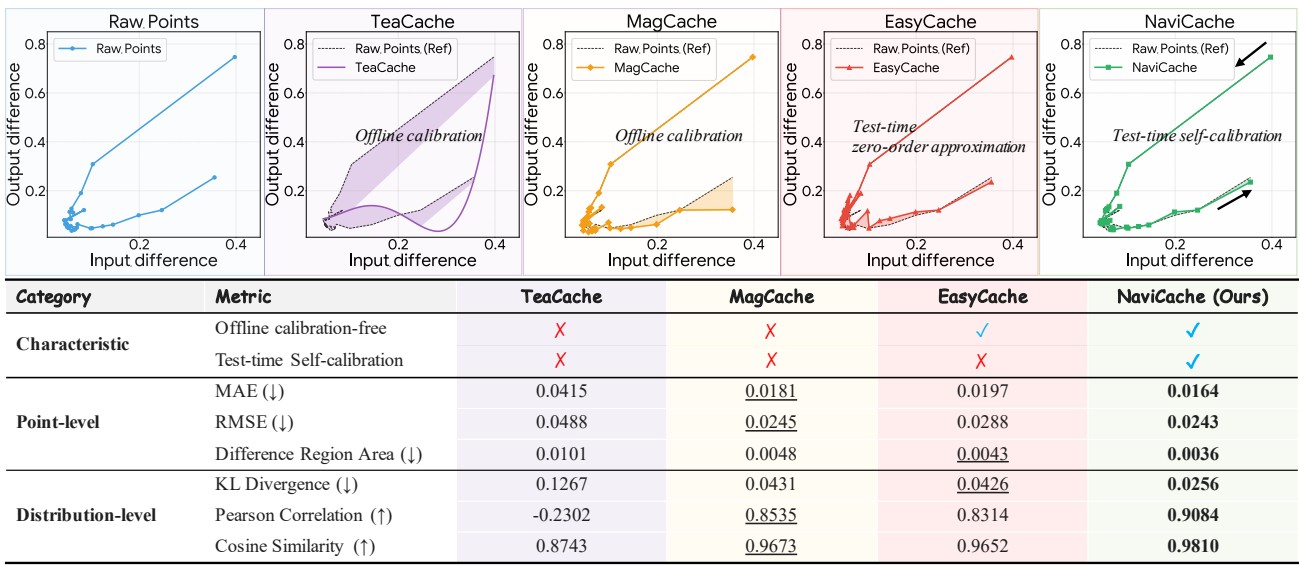

*Figure 1.* Comparison of prediction accuracy for determining whether to skip the output computation at a denoising step in VDMs. We visualize the relationship between input and output differences as a 2D manifold. The shaded areas are constructed by the segments connecting predicted coordinates and ground-truth coordinates (Raw Points). Based on quantitative evaluation across all prompts in VBench (Huang et al., 2024), we employ Point-level metrics (MAE, RMSE, and Difference Region Area) to measure instantaneous error, and Distribution-level metrics (KL Divergence, Pearson Correlation, and Cosine Similarity) to assess the alignment between predicted and real coordinates. *TeaCache* (Liu et al., 2025) and *MagCache* (Ma et al., 2026) rely on offline calibration, failing to capture runtime dynamics; *EasyCache* (Zhou et al., 2025) uses a test-time zero-order approximation which leads to significant lag. In contrast, our *NaviCache* achieves plug-and-play test-time self-calibration, accurately tracking the trajectory with minimal deviation.

challenges: *(i) Domain Variable Mismatch:* the two fields lack direct mapping; physical navigation variables like velocity or coordinates are absent in latent space, while video-specific metrics like feature drift are missing from classical control frameworks. *(ii) Non-Stationary Dynamics:* the diffusion process exhibits volatile dynamics in the initial and final stages, creating "high-turbulence" zones where standard estimation models tend to diverge.

To overcome these obstacles and facilitate efficient VDM inference, we present **NaviCache**, a plug-and-play test-time self-calibration framework introducing Inertial Navigation System (INS) principles tailored for diffusion dynamics. Rather than estimating absolute changes, *NaviCache* leverages the task-specific observability of input differences to perform dynamic correction of the output-to-input change ratio. To handle high-variance regimes at trajectory boundaries, *NaviCache* employs a strategic Initial Alignment, mandating full computation in initial steps to calibrate the system's state and uncertainty covariance. During the main diffusion phase, a dual-state estimation engine fuses noisy observations with predicted momentum, including Prior Projection and Observational Rectification. A prediction error threshold serves as a Measurement Update: whenever estimated uncertainty exceeds the bound, the system triggers a full computation to re-anchor the trajectory. We provide a theoretical guarantee that *NaviCache* achieves a lower estimation error bound than zero-order heuristics, ensuring higher fidelity under aggressive acceleration. As shown in Figure 1, our method for determining whether to skip the

output of a certain step in VDM exhibits greater accuracy in error judgment. As shown in Figure 1, our *NaviCache* exhibits significantly greater accuracy in error judgment.

Extensive experiments based on models from the *Wan*, *HunyuanVideo*, and *Open-Sora* series demonstrate that *NaviCache* significantly outperforms baselines.

Our contributions are summarized as follows:

- We reformulate VDM feature evolution as a state-space model, establishing the first formal mapping between diffusion dynamics and INS theory.

- We propose *NaviCache*, an offline calibration-free framework featuring a dual-state engine and initial alignment to accurately track feature manifolds.

- We theoretically prove that *NaviCache* achieves lower error than the zero-order heuristics used in existing calibration-free methods.

- Extensive experiments on *Wan*, *HunyuanVideo*, and *Open-Sora* demonstrate that *NaviCache* consistently outperforms baselines.

## 2. Related Work

**Video Diffusion Models.** Video diffusion models (VDMs) (Ho et al., 2020; Blattmann et al., 2023; Kuaishou Technology, 2024) have established a dominant paradigm for high-fidelity content generation. Initially, video generation relied on U-Net architectures extending image diffusion

priors (Ho et al., 2022; Blattmann et al., 2023). However, the field has undergone a transition toward Diffusion Transformers (DiTs) (Peebles & Xie, 2023) due to their superior scalability in modeling complex spatiotemporal dependencies. This shift has facilitated the emergence of several groundbreaking models: *HunyuanVideo* (Kong et al., 2024), which introduces a large-scale systematic production pipeline. *Open-Sora 1.2/2.0* (Zheng et al., 2024; 2025), focusing on democratizing efficient video creation. *Wan 2.1* (Wan Team et al., 2025) has demonstrated impressive performance in modeling intricate spatiotemporal relationships. Despite these advancements, the iterative denoising process in DiTs remains computationally expensive, presenting a significant barrier for real-time applications and necessitating efficient inference strategies.

**Video Generation Acceleration.** Video generation acceleration is gaining increasing attention (Guo et al., 2024; Xi et al., 2025; Liu et al., 2026). Methods like (Salimans & Ho, 2022; Meng et al., 2023) and Latent Consistency Models (Luo et al., 2023) reduce sampling steps through intensive retraining, but they suffer from prohibitive training costs and limited generalization across diverse prompts. To avoid model retraining, Training-free methods leverage feature redundancy to do acceleration: *PAB* (Zhao et al., 2025) reducing the quadratic complexity of attention mechanisms by broadcasting attention maps across frames, exploiting spatial-temporal redundancy in video diffusion. *MagCache* (Ma et al., 2026) introduces a magnitude-aware cache that skips unimportant diffusion timesteps, accelerating video generation with minimal quality loss. However, despite being training-free, these methods rely on offline calibration, which makes them difficult to apply universally across different models and datasets. Therefore, offline calibration-free methods, such as *EasyCache* (Zhou et al., 2025), propose an offline calibration-free, runtime-adaptive cache that reuses prior computations during inference to speed up VDMs. However, these methods typically rely on instantaneous zero-order approximations of feature dynamics, failing to capture the intrinsic momentum of the diffusion trajectory.

**Our Distinction.** *NaviCache* is an offline calibration-free method, distinguishing itself from *TeaCache* and *MagCache*. Unlike *EasyCache*, which applies zero-order approximations in offline calibration-free approaches, *NaviCache* reconceptualizes feature evolution as a kinematic navigation problem, enabling test-time self-calibration.

## 3. Methodology

### 3.1. Problem Formulation and Preliminary Analysis

Consider the denoising process of a VDM. We use $t$ to index denoising iterations in execution order. Let $\mathcal{F} : \mathbb{R}^d \to \mathbb{R}^d$

denote the exact transformation whose output may be cached during denoising. The computational cost is dominated by executing $\boldsymbol{y}_t = \mathcal{F}(\boldsymbol{x}_t)$, where $\boldsymbol{x}_t$ and $\boldsymbol{y}_t$ denote the input and output features at denoising iteration $t$, respectively. To accelerate inference, we aim to skip the execution of $\mathcal{F}$ if the output variation $\Delta \boldsymbol{O}_t = \|\boldsymbol{y}_t - \boldsymbol{y}_{t-\delta}\|_1$ is negligible, given the input variation $\Delta \boldsymbol{I}_t = \|\boldsymbol{x}_t - \boldsymbol{x}_{t-\delta}\|_1$, where $\delta \geq 1$ denotes the iteration offset to the reference feature used to measure feature variations.

**Offline calibration-based Methods.** Methods such as *TeaCache* (Liu et al., 2025) assume a global functional relationship between input and output differences. They approximate the output variation using a mapping function (e.g. polynomial function) parameterized by $\theta$:

$$\Delta \boldsymbol{O}_t \approx \mathcal{G}(\Delta \boldsymbol{I}_t; \theta^*) = \sum_{k=1}^{K} w_k (\Delta \boldsymbol{I}_t)^k + b, \qquad (1)$$

where $\theta^* = \{w_k, b\}$ are hyperparameters optimized on a calibration dataset $\mathcal{D}_{\text{cal}}$ to minimize the regression error $\mathbb{E}_{\mathcal{D}_{\text{cal}}}[\|\Delta \boldsymbol{O} - \mathcal{G}(\Delta \boldsymbol{I})\|]$. As illustrated in Figure 1, this paradigm suffers from three fundamental flaws: (1) *Dependency on Calibration Data:* Performance is strictly bound to the quality of $\mathcal{D}_{\text{cal}}$; (2) *Calibration Overhead:* The search for $\theta^*$ consumes significant GPU hours; (3) *Distribution Shift:* Since $\mathcal{D}_{\text{cal}} \neq \mathcal{D}_{\text{target}}$, the learned mapping $\mathcal{G}(\cdot; \theta^*)$ often generalizes poorly to unseen domains.

**Offline calibration-free Methods.** To circumvent calibration, offline calibration-free approaches like *EasyCache* adopt an instantaneous approximation. They assume the mapping function (e.g. the change ratio) observed at step $t-1$ remains constant at step $t$. This is formulated as:

$$\Delta \boldsymbol{O}_t \approx \phi_{t-1} \cdot \Delta \boldsymbol{I}_t, \quad \text{where} \quad \phi_{t-1} = \frac{\Delta \boldsymbol{O}_{t-1}}{\Delta \boldsymbol{I}_{t-1}}. \quad (2)$$

$\phi_t$ represents the instantaneous output-to-input sensitivity at timestep $t$. By reusing $\phi_{t-1}$, these methods implicitly apply a Zero-Order Hold (ZOH) assumption on the system dynamics: $\phi_t \approx \phi_{t-1}$. While this avoids distribution shift, the ZOH assumption fails to capture the *momentum* of feature evolution. In high-turbulence regions, $|\phi_t - \phi_{t-1}|$ is large, leading to estimation error and resulting in either visual artifacts (over-skipping) or insufficient acceleration (under-skipping).

### 3.2. *NaviCache*: Test-Time Self-Calibration Caching

As shown in Figure 2, we propose *NaviCache*, which reconceptualizes the estimation of $\phi_t$ as a state tracking problem within a Linear Gaussian State-Space Model (LGSSM). Unlike previous heuristics, *NaviCache* explicitly models the

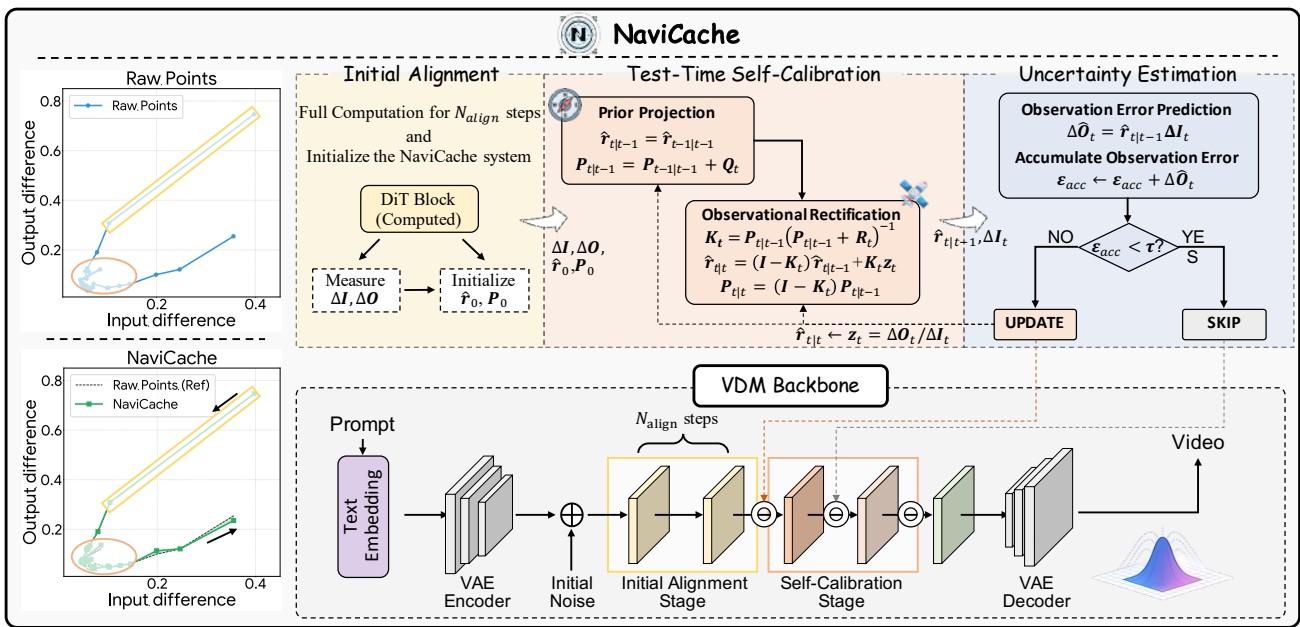

*Figure 2.* Overview of the *NaviCache* Framework. *NaviCache* reformulates feature caching as a recursive state-space tracking problem to enable offline calibration-free acceleration for VDMs. Left: Comparison between raw trajectories (top) and *NaviCache* predicted trajectories (bottom), categorized into the Initial Alignment Stage and the Self-Calibration Stage. Right: Detailed modular workflow. The former stage is dedicated to initializing the *NaviCache* system through full computation; the latter employs the Test-Time Self-Calibration engine to recursively track feature dynamics, integrated with an Uncertainty Estimation module to adaptively determine whether to SKIP the current execution. *NaviCache* ensures the VDM Backbone maintains high visual fidelity while significantly reducing cost.

uncertainty and the kinematic inertia of the feature evolution ratio. $r_t \in \mathbb{R}$ denotes the hidden state corresponding to the true sensitivity ratio $\frac{\Delta O_t}{\Delta I_t}$ at step $t$. We aim to derive an optimal estimator $\hat{r}_t$ that minimizes the mean squared estimation error.

### 3.2.1. INITIAL ALIGNMENT

In INS, a stationary alignment phase is strictly required to determine the initial attitude and biases before dynamic navigation begins. Similarly, the diffusion process exhibits non-stationary, chaotic dynamics in the early timesteps (high noise levels), where the ratio $r_t$ fluctuates violently. Standard recursive estimators may diverge if initialized in this regime without ground truth.

We introduce a strategic Initial Alignment phase for the first $N_{\text{align}}$ steps. During this phase, computation skipping is disabled. We compute the exact $\Delta O_k$ and $\Delta I_k$ for $k = 1, \ldots, N_{\text{align}}$ to construct an unbiased initialization for the state mean $\hat{r}_0$ and the error covariance $P_0$:

$$\hat{r}_0 = \frac{1}{N_{\text{align}}} \sum_{k=1}^{N_{\text{align}}} \frac{\Delta O_k}{\Delta I_k}, \quad P_0 = \sigma_{init}^2, \quad (3)$$

where $\sigma_{init}^2$ reflects the initial uncertainty. This ensures the estimator engages the "navigation mode" with a calibrated understanding of the feature manifold.

### 3.2.2. TEST-TIME SELF-CALIBRATION

To capture the temporal continuity of the ratio $r_t$, we employ a recursive Bayesian filter that tracks two sufficient statistics: the state estimate $\hat{r}_t$ and the estimation uncertainty $P_t$. This dual-state tracking allows *NaviCache* to be "self-aware" of its prediction reliability.

We define the system dynamics through the following stochastic differential equations (Process and Measurement Model), discretized for the timestep $t$:

$$r_t = F_t r_{t-1} + w_{t-1}, \qquad w_{t-1} \sim \mathcal{N}(0, Q_t) \quad (4)$$
$$z_t = H_t r_t + v_t, \qquad v_t \sim \mathcal{N}(0, R_t) \quad (5)$$

where $z_t = \Delta O_t / \Delta I_t$ is the observed ratio (available only during full computation), $Q_t$ represents the process noise covariance (modeling the intrinsic drift or "inertia" of the ratio), and $R_t$ denotes the measurement noise covariance (modeling the observational turbulence). In our implementation, we assume a random walk model for the ratio evolution ($F_t = I, H_t = I$).

The recursive estimation proceeds in two stages:

*Stage 1. Prior Projection.* Based on the posterior at $t-1$, we project the state and uncertainty forward to step $t$. This prediction incorporates the process noise $Q_t$, effectively modeling the system's momentum:

$$\hat{r}_{t|t-1} = \hat{r}_{t-1|t-1}, \quad P_{t|t-1} = P_{t-1|t-1} + Q_t. \quad (6)$$

Here, $P_{t|t-1}$ grows, reflecting increased uncertainty due to the potential drift of the ratio over time.

*Stage 2. Observational Rectification.* When a ground-truth measurement $z_t$ is available (triggered by the uncertainty threshold), our objective is to rectify the inertial prediction. Instead of a heuristic update, we formulate this as a *variance minimization problem*. We seek an optimal estimator $\hat{r}_{t|t}$ that linearly fuses the prediction and the new observation:

$$\hat{r}_{t|t} = (I - K_t)\hat{r}_{t|t-1} + K_t z_t, \tag{7}$$

where $K_t$ serves as the *Calibrated Fusion Factor (CFF)*. The posterior estimation uncertainty $P_{t|t}$ is derived as:

$$P_{t|t} = (I - K_t)^2 P_{t|t-1} + K_t^2 R_t. \tag{8}$$

To ensure optimality, we determine $K_t$ by minimizing the posterior uncertainty (setting the gradient $\frac{\partial P_{t|t}}{\partial K_t} = 0$):

$$\frac{\partial P_{t|t}}{\partial K_t} = -2(I - K_t)P_{t|t-1} + 2K_t R_t = 0. \tag{9}$$

Solving this yields the closed-form solution for the *Optimal CFF* (formally equivalent to the optimal gain):

$$K_t = P_{t|t-1}(P_{t|t-1} + R_t)^{-1}, \tag{10}$$
$$P_{t|t} = (I - K_t)P_{t|t-1}. \tag{11}$$

This mechanism acts as an intelligent gatekeeper: in stable regimes ($P_{t|t-1} \ll R_t$), $K_t \to 0$, prioritizing historical inertia; in turbulent regimes ($P_{t|t-1} \gg R_t$), $K_t \to I$, instantly aligning with the new observation to prevent error accumulation.

### 3.2.3. UNCERTAINTY ESTIMATION

The decision to execute or skip the current computation is governed by an Uncertainty-Aware Mechanism. We define an accumulated error metric $\mathcal{E}_{acc}$ that integrates the predicted output variation. For a step $t$, we predict the output variation as $\Delta \hat{O}_t = \hat{r}_{t|t-1} \cdot \Delta I_t$ and update $\mathcal{E}_{acc} \leftarrow \mathcal{E}_{acc} + \Delta \hat{O}_t$.

The system operates under a Safety Gate logic:

$$\text{Action}_t = \begin{cases} \textbf{SKIP} & \text{if } \mathcal{E}_{acc} < \tau \\ \textbf{UPDATE} & \text{if } \mathcal{E}_{acc} \geq \tau \end{cases} \tag{12}$$

where $\tau$ is the fidelity threshold.

- **SKIP** (Dead Reckoning): We bypass the current computation, utilizing the cached feature. The estimator continues to rely on the prior prediction.

- **UPDATE** (Measurement Fix): The accumulated error breaches the tolerance. We force a full current computation to obtain the true $\Delta O_t$, deriving the observation $z_t$. This triggers the A Posterior Calibration (Eq. 7-11), re-anchoring the state $r_t$ and collapsing the uncertainty variance $P_t$, $\mathcal{E}_{acc}$ is then reset to $\mathbf{0}$.

### 3.3. Theoretical Analysis

We provide a theoretical guarantee that *NaviCache* achieves a lower estimation error bound than zero-order heuristics. We adopt the MSE of the ratio estimation as our metric: $\mathcal{L}_t = \mathbb{E}[(\hat{r}_t - r_t)^2]$.

**Assumption 3.1** (Markovian Dynamics)**.** The evolution of the feature change ratio $r_t$ follows a Gauss-Markov process with process noise variance $Q$.

**Theorem 3.2** (Lower Error Bound of *NaviCache*)**.** *Consider the steady-state estimation where the CFF converges to a constant $K_\infty$. The posterior estimation error variance of NaviCache, denoted as $P_\infty$, satisfies the Algebraic Riccati Equation (ARE). For any zero-order heuristic that uses the previous observation $z_{t-1}$ as the estimate (i.e., Easy-Cache), the estimation error variance is $\sigma_{ZOH}^2 = Q + R$. NaviCache guarantees:*

$$P_\infty < \sigma_{ZOH}^2, \tag{13}$$

*provided $R > 0$ and $Q > 0$.*

*Remark* 3.3. Theorem 3.2 implies that by optimally weighting the historical prior and the current observation, *NaviCache* systematically reduces the variance of the prediction error compared to directly using the previous observation (which assumes $K_t = I$). The proof is provided in Appendix A.1.

## 4. Experiment

### 4.1. Experimental Setup

**Model selection.** We evaluate *NaviCache* on three VDMs: *HunyuanVideo* (Kong et al., 2024), *Wan2.1-1.3B* (Wan Team et al., 2025), *OpenSora 1.2* (Zheng et al., 2024). We compare with video generation acceleration methods including some straightforward methods, *PAB* (Zhao et al., 2025), *Tea-Cache* (Liu et al., 2025), *MagCache* (Ma et al., 2026), and *EasyCache* (Zhou et al., 2025) on *VBench* (Huang et al., 2024) dataset.

**Evaluation Metrics.** We follow *PAB* (Zhao et al., 2025), *MagCache* (Ma et al., 2026), and *EasyCache* (Zhou et al., 2025) *TeaCache* (Liu et al., 2025), focusing on two primary aspects: inference efficiency and visual quality to assess the performance of video generation acceleration methods. For evaluating inference efficiency, we use inference latency and speedup as metrics. For visual quality evaluation, we em-

*Table 1.* Comparison of efficiency and visual retention across different video generation models.

| Model | Reference | Efficiency | | Visual Retention | | | Vbench (%) ↑ |
|---|---|---|---|---|---|---|---|
| | | Latency(s) ↓ | Speedup ↑ | PSNR ↑ | SSIM ↑ | LPIPS ↓ | |
| *Wan 2.1-1.3B (81 frames, $832 \times 480$)* | | | | | | | |
| Wan 2.1 ($T = 50$) | - | 214.93 | 1.00× | - | - | - | 80.86 |
| + 40% steps | - | 100.37 | 2.14× | 14.50 | 0.5226 | 0.4374 | 80.30 |
| + Random 0.4 | - | 102.75 | 2.09× | 11.92 | 0.4204 | 0.5911 | 78.68 |
| + Static cache | - | 102.14 | 2.10× | 14.18 | 0.5007 | 0.4789 | 79.58 |
| + PAB (Zhao et al., 2025) | ICLR'25 | 141.28 | 1.52× | 18.84 | 0.6484 | 0.3010 | 77.60 |
| + TeaCache (Liu et al., 2025) | CVPR'25 | 121.57 | 1.77× | 22.79 | 0.8169 | 0.0952 | **80.67** |
| + MagCache (Ma et al., 2026) | NeurIPS'25 | 105.05 | 2.05× | 23.33 | 0.8331 | 0.0958 | 80.26 |
| + EasyCache (Zhou et al., 2025) | - | 99.21 | 2.17× | 22.96 | 0.7935 | 0.1053 | 80.18 |
| *+ NaviCache (Ours) - fast* | - | 96.40 | 2.23× | 23.46 | 0.8011 | 0.0956 | 80.21 |
| *+ NaviCache (Ours) - mid* | - | 106.97 | 2.01× | 24.08 | 0.8333 | 0.0839 | 80.38 |
| *+ NaviCache (Ours) - slow* | - | 115.86 | 1.86× | **25.10** | **0.8638** | **0.0686** | 80.58 |
| *HunyuanVideo (129 frames, $960 \times 544$)* | | | | | | | |
| HunyuanVideo ($T = 50$) | - | 2363.83 | 1.00× | - | - | - | 82.53 |
| + 50% steps | - | 1203.44 | 1.96× | 18.79 | 0.7101 | 0.3319 | 81.78 |
| + Random 0.5 | - | 1213.09 | 1.95× | 19.85 | 0.7201 | 0.3214 | 81.04 |
| + Static cache | - | 1338.51 | 1.77× | 18.74 | 0.7081 | 0.3309 | 81.76 |
| + PAB (Zhao et al., 2025) | ICLR'25 | 1700.60 | 1.39× | 18.58 | 0.7023 | 0.3827 | 76.98 |
| + TeaCache (Liu et al., 2025) | CVPR'25 | 1070.14 | 2.21× | 24.05 | 0.8046 | 0.1830 | **82.32** |
| + MagCache (Ma et al., 2026) | NeurIPS'25 | 882.76 | 2.68× | 29.83 | 0.8904 | 0.0941 | 81.99 |
| + EasyCache (Zhou et al., 2025) | - | 1100.30 | 2.15× | 32.53 | 0.9241 | 0.0589 | 82.04 |
| *+ NaviCache (Ours) - fast* | - | 928.45 | 2.55× | 30.64 | 0.8982 | 0.0860 | 82.03 |
| *+ NaviCache (Ours) - mid* | - | 1089.43 | 2.17× | 32.65 | 0.9256 | 0.0571 | 82.07 |
| *+ NaviCache (Ours) - slow* | - | 1150.87 | 2.05× | **33.87** | **0.9357** | **0.0462** | 82.15 |
| *Open-Sora 1.2 (51 frames, $848 \times 480$)* | | | | | | | |
| Open-Sora 1.2 ($T = 30$) | - | 56.48 | 1.00× | - | - | - | 79.25 |
| + 50% steps | - | 29.84 | 1.89× | 15.82 | 0.6961 | 0.3363 | 77.36 |
| + Random 0.5 | - | 29.71 | 1.90× | 16.51 | 0.7037 | 0.3264 | 76.78 |
| + Static cache | - | 30.47 | 1.85× | 15.73 | 0.6961 | 0.3382 | 77.37 |
| + PAB (Zhao et al., 2025) | ICLR'25 | 45.51 | 1.24× | 23.58 | 0.8220 | 0.1743 | 76.95 |
| + TeaCache (Liu et al., 2025) | CVPR'25 | 41.38 | 1.36× | 23.16 | 0.8335 | 0.1429 | **79.10** |
| + MagCache (Ma et al., 2026) | NeurIPS'25 | 26.07 | 2.17× | 22.33 | 0.8207 | 0.1676 | 77.97 |
| + EasyCache (Zhou et al., 2025) | - | 34.55 | 1.63× | 23.63 | 0.8458 | 0.1320 | 78.83 |
| *+ NaviCache (Ours) - fast* | - | 31.80 | 1.78× | 22.53 | 0.8255 | 0.1551 | 79.05 |
| *+ NaviCache (Ours) - mid* | - | 35.29 | 1.60× | 23.67 | 0.8463 | 0.1309 | 79.03 |
| *+ NaviCache (Ours) - slow* | - | 40.98 | 1.38× | **26.37** | **0.8860** | **0.0917** | 79.08 |

ploy LPIPS (Zhang et al., 2018), PSNR, SSIM and *VBench* (Huang et al., 2024) score.

**Implementation Detail.** We utilize *NaviCache* to dynamically estimate the sensitivity-ratio state $\hat{r}_t$. To validate the robustness of the algorithm, we maintain consistent filter parameter settings across all models (*Wan 2.1, HunyuanVideo,* and *Open-Sora 1.2*). Specifically, the state estimate $\hat{r}_0$ is initialized as a zero matrix, and the error covariance $P$ is initialized as an identity matrix to accommodate uncertainty during the initial inference phase. Regarding the noise parameters, we balance estimation sensitivity and stability by setting both the Process Noise Covariance $Q$ and the Measurement Noise Covariance $R$ to

$0.05I$, where $I$ denotes the identity matrix. To evaluate *NaviCache*, we employ specific Initial Alignment steps ($N_{\text{align}}$) and error thresholds ($\tau$) across three modes: fast, mid, and slow. Specifically, $N_{\text{align}}$ is set to 5 for *HunyuanVideo* and *Open-Sora 1.2*, and 10 for *Wan 2.1*. The error thresholds $\tau$ are adaptively configured for each model and mode: for *HunyuanVideo*, $\tau \in \{0.040I, 0.035I, 0.025I\}$; for *Wan 2.1*, $\tau \in \{0.07I, 0.05I, 0.04I\}$; and for *Open-Sora 1.2*, $\tau \in \{0.55I, 0.35I, 0.15I\}$.

### 4.2. Main Results

Table 1 presents a comprehensive comparison of *NaviCache* against state-of-the-art acceleration methods across three

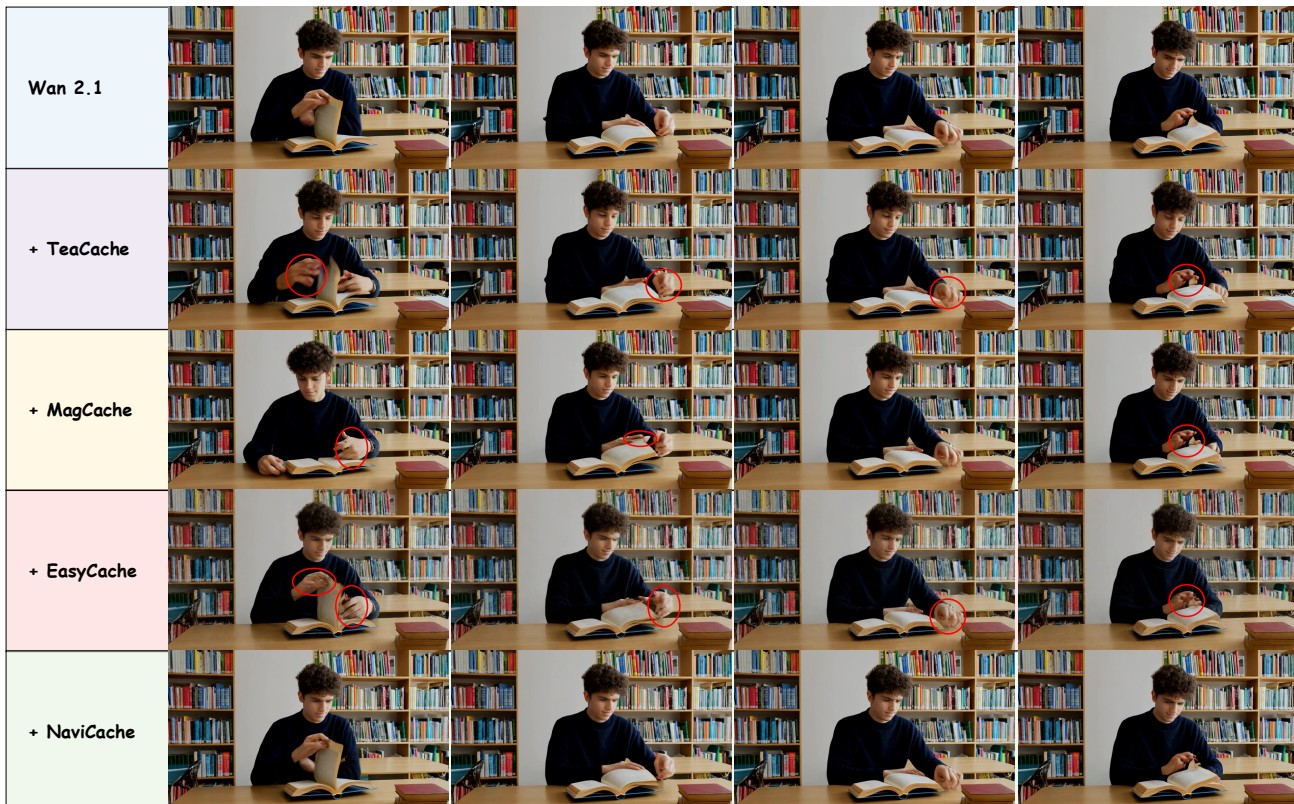

*Figure 3.* Visualization and Case Study of video generation. For the prompt "*A young man sitting at a desk in a library reading*", compared to existing methods that suffer from blurriness and structural artifacts (highlighted in red), our *NaviCache* preserves high visual fidelity and motion continuity, closely matching the unaccelerated baseline.

representative VDMs: *Wan 2.1*, *HunyuanVideo*, and *Open-Sora 1.2*. To accommodate diverse deployment requirements and hardware constraints, we provide three distinct configurations: fast, mid, and slow.

When comparing methods within similar latency envelopes, *NaviCache* consistently demonstrates superior visual retention over existing offline calibration-free methods. For instance, on *HunyuanVideo*, while *EasyCache* achieves a 2.15× speedup with a PSNR of 32.53, our mid mode provides a higher 2.17× speedup while improving PSNR to 32.65 and reducing LPIPS to 0.0571. This advantage stems from our test-time self-calibration engine, which captures the kinematic momentum of feature evolution more accurately than the calibration-free, zero-order hold baselines.

In comparison to offline calibration-based methods, *NaviCache* exhibits remarkable fidelity. Across all three backbone models, our slow mode achieves PSNR, SSIM, and LPIPS results that significantly outperform *TeaCache*, MagCache, and PAB. On *Wan 2.1*, *NaviCache*-slow reaches a PSNR of 25.10, substantially higher than *TeaCache* (22.79) and MagCache (23.33). Although *TeaCache* maintains a slight lead in Vbench scores in some cases—likely due to its prompt-specific polynomial fitting on calibration sets—*NaviCache* remains highly competitive, with its

Vbench scores staying within a marginal gap while offering drastically better pixel-level and perceptual reconstruction. More importantly, as demonstrated in the subsequent "Visualization and Case Study" section, despite *TeaCache*'s marginal advantage in VBench scores, its actual generation quality is often suboptimal compared to *NaviCache*. These results validate that modeling the diffusion process as a self-calibrating navigation system effectively bridges the performance gap between calibration-free flexibility and calibration-dependent accuracy.

### 4.3. Visualization and Case Study

To further evaluate the visual fidelity of the accelerated videos, we present a case study using the *Wan 2.1* model with the prompt: "*A young man sitting at a desk in a library reading*". As illustrated in Figure 3, the unaccelerated *Wan 2.1* model demonstrates a robust understanding of the scene, generating smooth motion and consistent object details. Existing acceleration methods (e.g., *TeaCache*, *MagCache*, and *EasyCache*) often suffer from temporal inconsistencies, ghosting artifacts in high-motion areas, or semantic misalignment. Specifically, TeaCache and MagCache are prone to factual errors in objects, such as hands being separated or fingers deforming, and they suffer from occasional object

deformation and loss of fine textures during high-dynamic segments. *EasyCache*'s major issue is the severe visual artifacts, including significant motion blur and structural distortion in the subject's hands and the book pages. In contrast, our proposed *NaviCache* effectively mitigates these issues. By leveraging test-time self-calibration to track the feature evolution trajectory, *NaviCache* achieves superior visual fidelity, effectively maintaining structural integrity and high-frequency details even during rapid movement and complex object interactions. This visual evidence aligns with quantitative results in Table 1, confirming that *NaviCache* provides a better balance between eliminating calibration overhead and preserving high-quality video generation.

### 4.4. Ablation Study and Depth Analysis

**The impact of Initial Alignment steps $N_{\mathrm{align}}$.** The strategic Initial Alignment phase is critical for re-anchoring the navigation manifold in the high-turbulence initial stages of the diffusion process. We analyzed the impact of $N_{\mathrm{align}}$ on *NaviCache* across all models and observed consistent patterns. As shown in Table 2, we present the performance comparisons on *Wan 2.1* as a representative case. *(i)* Fidelity Enhancement: Increasing $N_{\mathrm{align}}$ from 5 to 15 steps (representing 10% to 30% of the total diffusion steps) yields a consistent improvement across all visual quality metrics. Specifically, PSNR increases from 22.73 dB to 24.31 dB, while LPIPS perceptual distance drops from 0.1095 to 0.0818. This confirms our hypothesis that a sufficient alignment duration allows the recursive Bayesian filter to build a highly accurate initial estimate of the state mean $\hat{r}_0$ and stabilize the uncertainty covariance $P_0$ before engaging the autonomous navigation mode. *(ii)* Efficiency Trade-off: While a larger $N_{\mathrm{align}}$ enhances structural coherence, it naturally increases inference latency due to fewer skipping opportunities in the early denoising stage. However, even with a conservative alignment of 15 steps ($N_{\mathrm{align}} = 15$), *NaviCache* maintains a competitive latency of 112.49s, demonstrating an efficient Pareto frontier compared to the unaccelerated baseline. *(iii)* Balanced Configuration: We select $N_{\mathrm{align}} = 10$ (20% ratio) as our default configuration for the *Wan 2.1*. This setting strikes an optimal balance, achieving superior visual retention (PSNR > 23 dB) while maintaining a high speedup ratio, effectively navigating the trade-off between the non-stationary dynamics of the initial steps and the computational gains of the diffusion phase.

*Table 2.* Ablation study on the Initial Alignment steps ($N_{align}$).

| $N_{align}$ | Ratio (%) | Latency (s) ↓ | PSNR ↑ | SSIM ↑ | LPIPS ↓ |
|---|---|---|---|---|---|
| 5 | 10% | 92.31 | 22.73 | 0.7786 | 0.1095 |
| 10 | 20% | 96.40 | 23.46 | 0.8011 | 0.0956 |
| 15 | 30% | 112.49 | 24.31 | 0.8191 | 0.0818 |

**The impact of state-space Covariances ($Q$, $R$) and threshold ($\tau$).** We analyzed the impact of Process Noise Covariance $Q$, Measurement Noise Covariance $R$, and the fidelity threshold $\tau$ on *NaviCache* across all models and observed consistent patterns. As demonstrated in Table 3, we present the performance comparisons on *Wan 2.1* as a representative case. These parameters collectively modulate the system's trade-off between estimation responsiveness and structural stability. *(i)* Process Noise $Q$: Decreasing $Q$ (from $0.05I$ to $0.005I$) significantly enhances visual fidelity across both $\tau$ regimes. For $\tau = 0.04I$, a smaller $Q$ improves PSNR from 25.10 dB to 30.33 dB. This aligns with our recursive dual-state engine logic: a lower $Q$ implies a more stable underlying feature evolution, leading the filter to rely more on the predicted momentum and reducing the frequency of erratic updates. However, this gain in quality comes at the cost of higher latency (e.g., 153.29s vs. 115.86s) due to a more conservative skipping strategy. *(ii)* Measurement Noise $R$: Conversely, a smaller $R$ (higher trust in new observations) tends to decrease latency but reduces visual retention. At $\tau = 0.07I$, reducing $R$ to $0.005I$ results in the lowest latency of 89.20s but sacrifices PSNR to 22.36 dB. This reflects the Observational Rectification stage (Eq. 7-11), where a high CFF causes the system to anchor too aggressively to noisy test-time observations, leading to cumulative drift in high-turbulence zones. *(iii)* Threshold $\tau$: The fidelity threshold $\tau$ acts as a global regulator of the Safety Gate logic. Increasing $\tau$ from $0.04I$ to $0.07I$ consistently boosts speedup but leads to an expected degradation in metrics such as LPIPS and SSIM. Our default choice ($Q = 0.05I$, $R = 0.05I$) provides a balanced Pareto frontier, achieving competitive visual fidelity (PSNR > 23 dB) while maintaining significant inference acceleration.

*Table 3.* Ablation of covariances ($Q$, $R$) and threshold ($\tau$).

| Hyperparameter | | | Latency(s) ↓ | Visual Retention | | |
|---|---|---|---|---|---|---|
| $\tau(I)$ | $Q(I)$ | $R(I)$ | | PSNR ↑ | SSIM ↑ | LPIPS ↓ |
| | 0.05 | 0.05 | 115.86 | 25.10 | 0.8638 | 0.0686 |
| 0.04 | 0.005 | 0.05 | 153.29 | 30.33 | 0.9458 | 0.0254 |
| | 0.05 | 0.005 | 108.10 | 24.92 | 0.8547 | 0.0711 |
| | 0.05 | 0.05 | 96.40 | 23.46 | 0.8011 | 0.0956 |
| 0.07 | 0.005 | 0.05 | 134.29 | 27.72 | 0.9128 | 0.0415 |
| | 0.05 | 0.005 | 89.20 | 22.36 | 0.7529 | 0.1202 |

**Adaptivity of the skipping strategy.** As illustrated in Figure 4, the step-skipping behavior of *NaviCache* exhibits a dynamically responsive pattern. This allows the system to capture the unique feature evolution of each prompt better, demonstrating a clear efficiency advantage over the relatively fixed computation allocation employed by calibration-dependent methods like TeaCache and MagCache.

**Scalability across Spatial-Temporal Configurations.** To evaluate the robustness of our method under varying computational loads, we analyze the inference latency across different video lengths and resolutions. As illustrated in Figure 5, we test spatial dimensions of 480p and 720p alongside

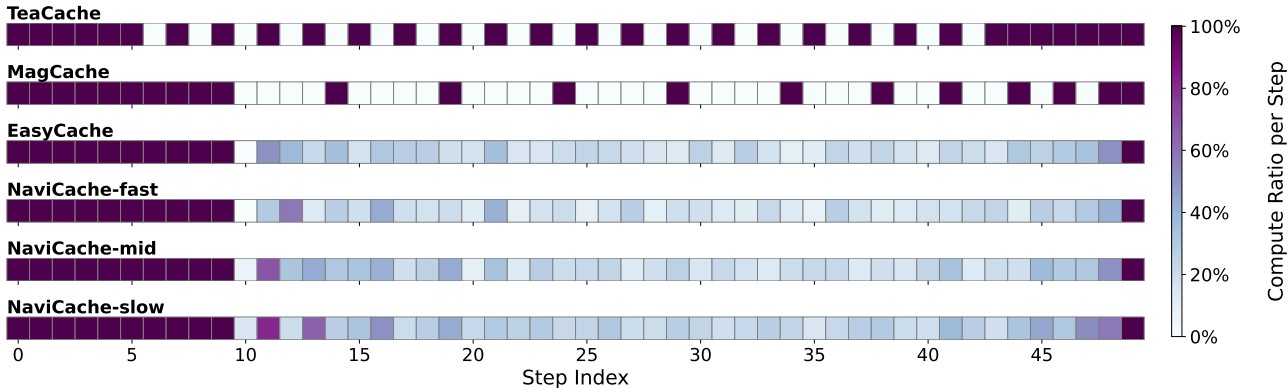

*Figure 4.* Comparison of skip frequency across timesteps based on Wan 2.1. The distribution illustrates how *NaviCache* adaptively allocates computation for individual samples.

temporal lengths of 51 and 102 frames using *Open-Sora 1.2*. The results demonstrate that *NaviCache* maintains highly stable acceleration performance, consistently achieving a speedup between $1.81\times$ and $1.91\times$. This confirms our test-time calibration framework scales efficiently without bottlenecks from increases in sequence length or pixel density.

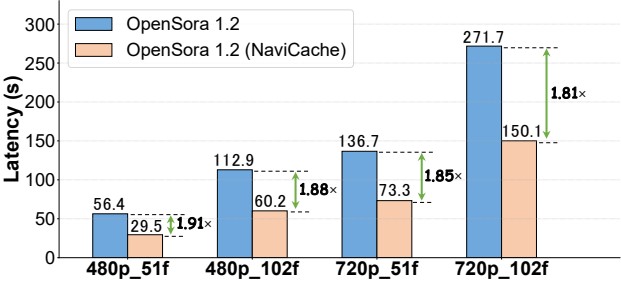

*Figure 5.* Inference latency under varying spatial and temporal configurations. Evaluations are conducted on 40 VBench prompts, with "p" and "f" denoting pixels and frames, respectively.

**Time Overhead and Calibration Cost.** As detailed in Table 4, we analyze the auxiliary time costs associated with different acceleration strategies. *NaviCache* requires zero offline calibration, fundamentally avoiding the risk of biased calibration data distributions that inherently constrain calibration-dependent baselines like TeaCache and Mag-Cache. Furthermore, the test-time computational latency of our state estimation algorithm (skipping justification) is merely 0.0109 seconds, entirely negligible relative to the substantial overall time saved during the diffusion process.

*Table 4.* Comparison of auxiliary time overhead. Measured on a single RTX 4090 GPU with batch size=1.

| Method | Offline Calibration (s) | Skipping Justification (s) |
|---|---|---|
| TeaCache | 15191 | 0.0142 |
| MagCache | 217 | 0.0028 |
| EasyCache | 0 | 0.0101 |
| NaviCache | 0 | 0.0109 |

## 5. Conclusion

We presented *NaviCache*, a test-time self-calibration framework that reformulates VDM feature evolution as an INS problem. By employing a dual-state estimation engine, *NaviCache* eliminates offline calibration overhead while achieving theoretically tighter error bounds than zero-order heuristics. Extensive experiments on *Wan*, *HunyuanVideo*, and *Open-Sora* confirm that *NaviCache* consistently delivers superior visual fidelity and more accurate error judgment for computation skipping. By bridging control theory and diffusion dynamics, *NaviCache* enables more efficient and principled deployment of large-scale generative models.

## Acknowledgment

National Natural Science Foundation of China (No. U24A20326, 62441236, 62402429, 62572423), the Key Research and Development Program of Zhejiang Province (No. 2025C01026), Ningbo Yongjiang Talent Introduction Programme (2023A-397-G), Young Elite Scientists Sponsorship Program by CAST (2024QNRC001). The author gratefully acknowledges the support of Zhejiang University Education Foundation Qizhen Scholar Foundation.

## Impact Statement

This work democratizes high-definition video generation by eliminating the reliance on costly calibration processes and specialized datasets. By enabling efficient, hardware-agnostic deployment through robust self-calibration, NaviCache significantly lowers the computational and energy barrier for real-time applications, contributing to the goal of Green AI. However, the reduced latency in video synthesis may also accelerate the production of deepfakes and misinformation. We advocate for the parallel development of detection algorithms and watermarking protocols to mitigate the risks associated with high-speed content generation.

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

# A. Appendix

## A.1. Theoretical Derivations

In this appendix, we provide the detailed derivation of the error bounds for the Zero-Order Hold (ZOH) method (e.g., *EasyCache*) and our proposed *NaviCache*, proving Theorem 3.2.

### A.1.1. System Model Definitions

We model the true ratio $r_t$ of the input-output variation as a discrete-time random walk process:

$$r_t = r_{t-1} + w_{t-1}, \quad w_{t-1} \sim \mathcal{N}(0, Q) \tag{14}$$

The observation $z_t$ (available when we do not skip) is noisy:

$$z_t = r_t + v_t, \quad v_t \sim \mathcal{N}(0, R) \tag{15}$$

where $Q$ is the process noise variance (representing the rate of change of the ratio) and $R$ is the measurement noise variance.

For clarity and generality, we use vector-matrix notation for derivations, as the state variables can be one-dimensional or multi-dimensional. When the state dimension $d = 1$, $r, z, P, Q, R, K, \tau, \sigma, \mathcal{E}, H, F, w, v$ will degenerate to scalars.

### A.1.2. Analysis of Zero-Order Heuristics (*EasyCache*)

Training-free methods like *EasyCache* approximate the current ratio $r_t$ using the most recent observation $z_{t-1}$ (assuming the step $t - 1$ was computed).

$$\hat{r}_t^{\text{ZOH}} = z_{t-1} \tag{16}$$

The estimation error $e_t^{\text{ZOH}} = \hat{r}_t^{\text{ZOH}} - r_t$ can be expanded as:

$$\begin{aligned} e_t^{\text{ZOH}} &= z_{t-1} - r_t \\ &= (r_{t-1} + v_{t-1}) - (r_{t-1} + w_{t-1}) \\ &= v_{t-1} - w_{t-1} \end{aligned} \tag{17}$$

Assuming the noise processes $v$ and $w$ are independent and zero-mean, the Mean Squared Error (MSE) is:

$$\begin{aligned} \sigma_{ZOH}^2 &= \mathbb{E}[(v_{t-1} - w_{t-1})^2] \\ &= \mathbb{E}[v_{t-1}^2] + \mathbb{E}[w_{t-1}^2] \\ &= R + Q \end{aligned} \tag{18}$$

This indicates that the error of zero-order heuristics is the direct sum of the measurement noise and the process drift.

### A.1.3. Analysis of *NaviCache* (Optimal Estimation)

Unlike heuristics that rely on a fixed trust assumption, *NaviCache* employs the recursive variance minimization derived in Eq. (11). We now analyze its theoretical performance in the steady state ($t \to \infty$).

Let the posterior estimation uncertainty converge to a constant value $P_\infty$, and the prior prediction uncertainty converge to $P_\infty^-$. Based on the evolution model, these relate as:

$$P_\infty^- = P_\infty + Q. \tag{19}$$

Substituting the optimal fusion factor $K_\infty = \frac{P_\infty^-}{P_\infty^- + R}$ into the posterior variance update Eq. (11), we obtain the steady-state variance equation:

$$P_\infty = (I - \frac{P_\infty^-}{P_\infty^- + R})P_\infty^- = \frac{RP_\infty^-}{P_\infty^- + R} = \frac{R(P_\infty + Q)}{P_\infty + Q + R}. \tag{20}$$

Rearranging this yields the quadratic convergence equation (formally known as the Algebraic Riccati Equation):

$$P_\infty^2 + QP_\infty - QR = 0. \tag{21}$$

Since variance must be non-negative, we take the positive root:

$$P_\infty = \frac{-Q + \sqrt{Q^2 + 4QR}}{2}. \tag{22}$$

### A.1.4. Proof of Inequality (Theorem 3.2)

We provide the proof that the estimation error of *NaviCache* is strictly bounded below that of zero-order heuristics. Our goal is to prove two properties:

1. The posterior estimation error is bounded by the sensor noise: $P_\infty < \sigma_{ZOH}^2$.

2. The prior prediction error (used during computation skipping) is bounded by the ZOH error: $P_\infty^- < \sigma_{ZOH}^2$.

**Part 1: Posterior Bound.** From the steady-state variance equation (Eq. 22), given $Q > 0$ and $R > 0$, to prove $P_\infty < \sigma_{ZOH}^2$, i.e., $P_\infty < Q + R$.

We show:

$$
\begin{aligned}
P_\infty &= \frac{-Q + \sqrt{Q^2 + 4QR}}{2} \\
&< \frac{-Q + \sqrt{Q^2 + 4QR + 4R^2}}{2} \quad \text{(Adding positive terms } 4R^2 \text{ inside sqrt)} \\
&= \frac{-Q + \sqrt{(Q + 2R)^2}}{2} \\
&= \frac{-Q + Q + 2R}{2} \\
&= R
\end{aligned}
\tag{23}
$$

Thus, we confirm $P_\infty < R < Q + R = \sigma_{\text{ZOH}}^2$.

**Part 2: Prior Bound (Computation Skipping Fidelity).** The Mean Squared Error of Zero-Order Hold (ZOH) heuristics is the sum of process drift and measurement noise: $\sigma_{ZOH}^2 = Q + R$. The steady-state prior variance of *NaviCache* is $P_\infty^- = P_\infty + Q$, indeed:

$$P_\infty^- = \frac{Q + \sqrt{Q^2 + 4QR}}{2} < Q + R, \tag{24}$$

From Eq. 23, we know $P_\infty < R$. Therefore:

$$P_\infty^- = P_\infty + Q < R + Q, \tag{25}$$

Substituting the definitions, we obtain:

$$P_\infty^- < \sigma_{ZOH}^2. \tag{26}$$

**Remark.** This inequality is critical for the acceleration mechanism. It theoretically guarantees that even when *NaviCache* *skips* the computation (relying solely on the inertial prediction $P_\infty^-$), its estimation error is still strictly lower than that of the baseline method ($Q + R$), ensuring higher fidelity during acceleration.

■

## A.2. Trajectory Analysis across Diverse Conditions

To further demonstrate the robustness of our premise, we visualize the feature evolution trajectories encompassing various models (*HunyuanVideo*, *Wan 2.1*, *Open-Sora 1.2*) and schedulers (e.g., UniPC, DPM++). As shown in Figure 6, across these diverse configurations, all trajectories consistently align with our core hypothesis: the ratio between output differences and input differences evolves smoothly along the diffusion process. The rate of feature change maintains distinct momentum and structural continuity, firmly verifying the broad applicability of our underlying theory.

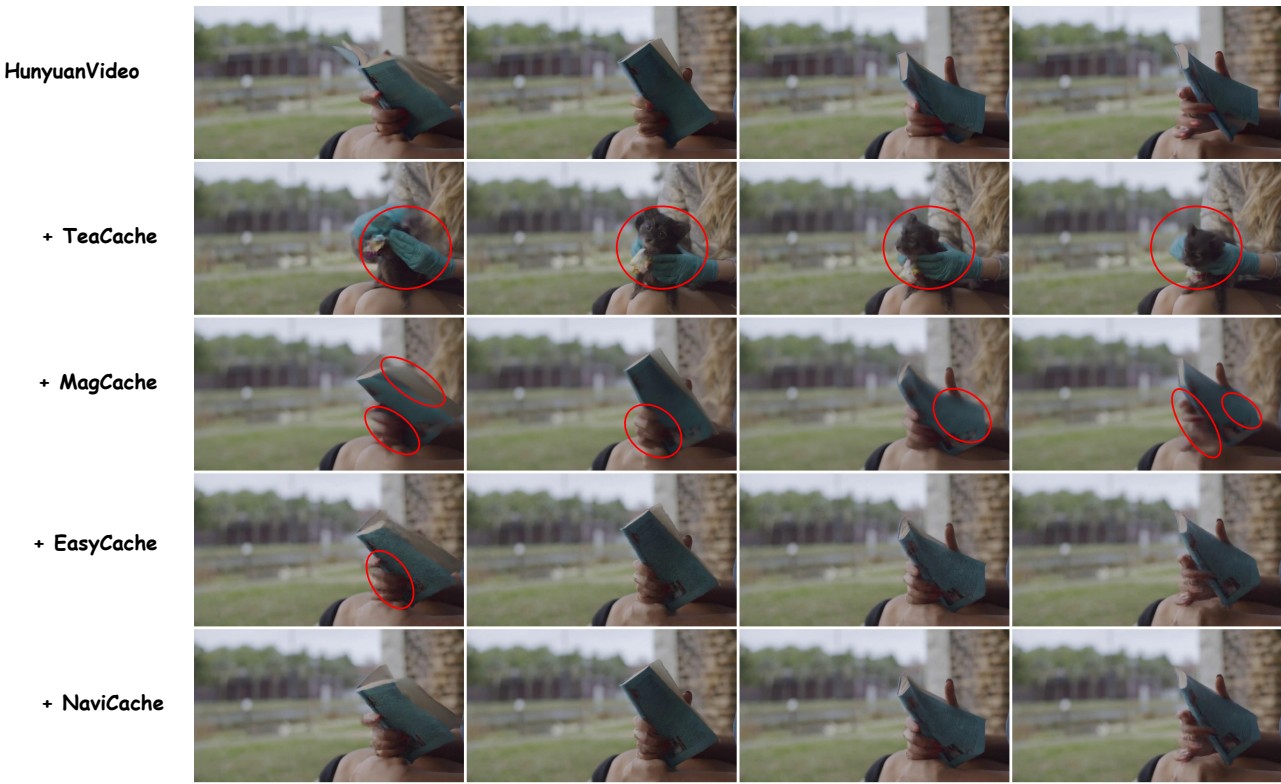

*Figure 6.* Trajectory visualization across various diffusion models and schedulers. Evaluated under diverse spatial-temporal configurations and ODE solvers to validate the consistent momentum of feature evolution. Solid lines represent the mean paths, while the shaded areas denote marginal variations.

## A.3. Supplementary Visualization and Case Study

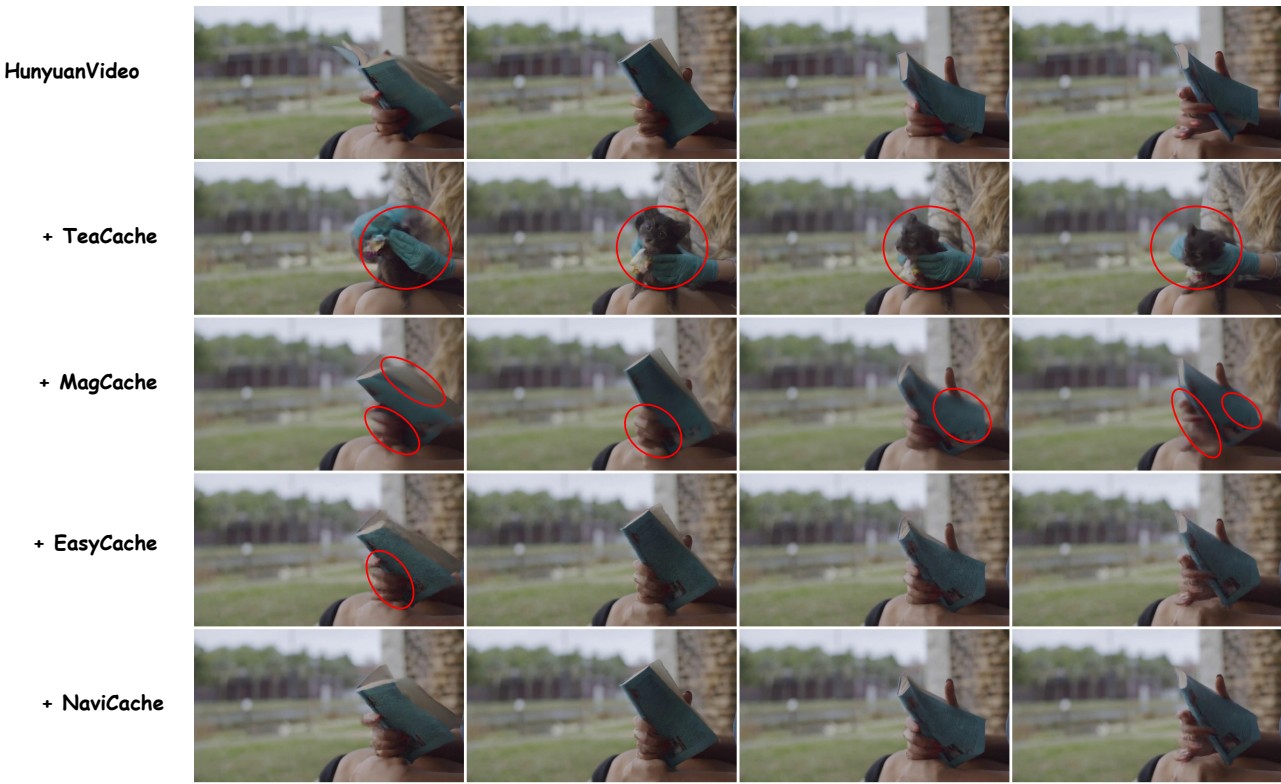

*Figure 7.* Qualitative comparison on HunyuanVideo using the prompt: "*a book*." TeaCache exhibits severe semantic misalignment, failing to adhere to the prompt by generating an unrelated subject (a dog). Furthermore, MagCache struggles with fine-grained morphological details, showing less distinct hand contours and pronounced ghosting artifacts during the page-turning motion compared to our method. Best viewed with zoom-in.

To further validate the robustness and visual quality of *NaviCache*, we provide additional qualitative comparisons in Figures 7 and 8. Consistent with the findings in the main text, existing acceleration methods exhibit various degrees of degradation.

As illustrated in Figure 7, *TeaCache* suffers from severe semantic misalignment, erroneously transforming the "book" into an unrelated "cat" (highlighted in red circles). While *MagCache* avoids such catastrophic semantic errors, it fails to maintain structural integrity, resulting in blurred hand contours and noticeable ghosting artifacts during the page-turning motion. Although *EasyCache* appears relatively stable in static frames, it introduces significant motion blur and visual artifacts throughout the sequence (can refer to the supplementary videos), failing to preserve fine-grained textures.

Similar patterns are observed in Figure 8, which depicts high-dynamic motion in a cyberpunk style. Both *TeaCache* and *MagCache* exhibit visible temporal inconsistencies and structural deformations around the astronaut's legs, leading to unnatural movements and blurred details (can refer to the supplementary videos). In contrast, *NaviCache* consistently

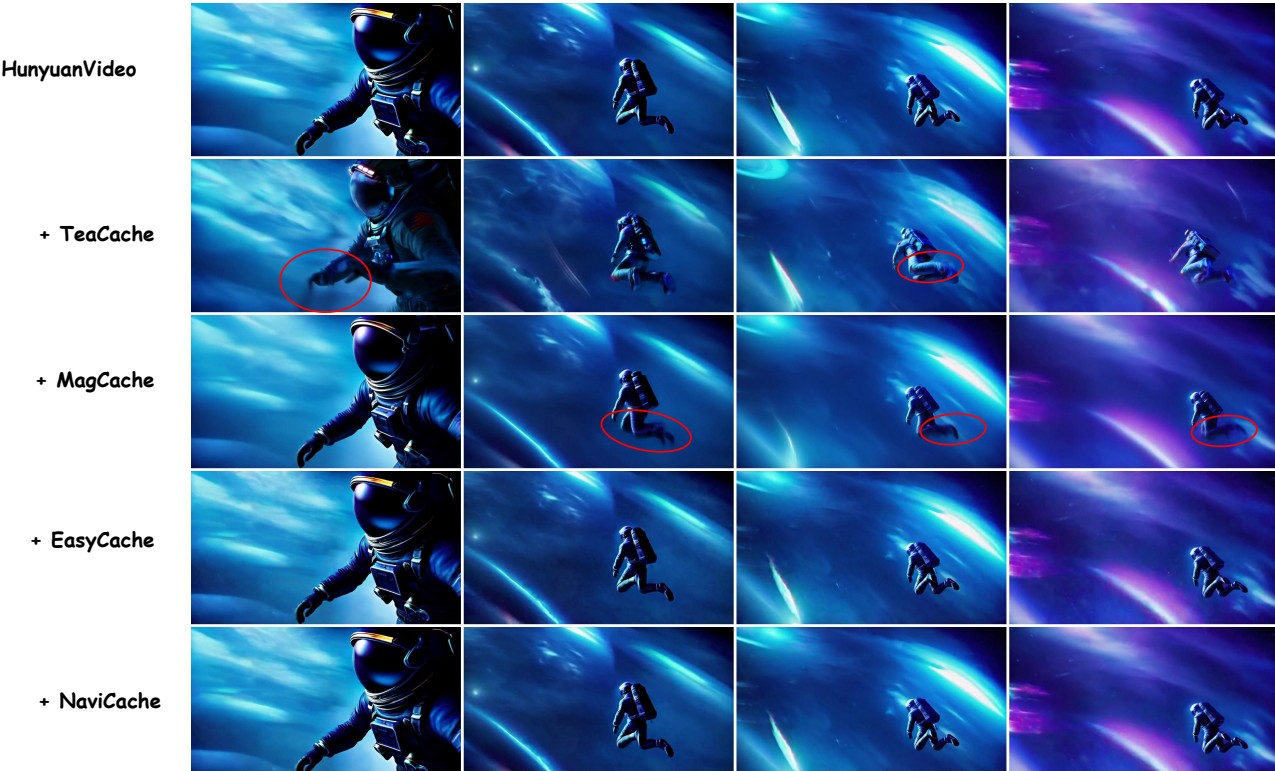

*Figure 8.* Qualitative comparison on HunyuanVideo using the prompt: "*An astronaut flying in space, in cyberpunk style.*" Notably, the outputs from MagCache and TeaCache exhibit visible ghosting artifacts and temporal inconsistencies around the astronaut's legs, whereas our method maintains clear structural integrity during motion. Best viewed with zoom-in.

generates high-fidelity videos with sharp details and precise motion trajectories. These cases demonstrate that by accurately tracking feature evolution through self-calibration, *NaviCache* effectively mitigates factual errors and artifacts, outperforming baseline methods in both semantic consistency and structural preservation.

