# OpenReview forum: "NaviCache: Test-Time Self-Calibration Caching for Video Generation"
_ICML.cc/2026/Conference — ICML 2026 regular_

### Official Review · Reviewer_csEP · 2026-03-11

**Soundness:** 3
**Presentation:** 3
**Significance:** 3
**Originality:** 3
**Overall Recommendation:** 4
**Confidence:** 3

**Summary:**

The paper proposes NaviCache, a plug-and-play, offline calibration-free test-time self-calibration method to accelerate video diffusion models (VDMs). The key idea is to reframe the mapping between input and output feature differences of a DiT block as a latent state x_t = ΔO_t/ΔI_t tracked by a linear Gaussian state-space model, enabling recursive estimation (akin to a Kalman filter) with an initial alignment phase, uncertainty-aware updates, and an error-bounded block-skipping policy. The authors provide a steady-state variance analysis showing a lower estimation error bound than zero-order hold heuristics and report improved speed–quality trade-offs over TeaCache, MagCache, and EasyCache on Hunyuan-Video, Wan 2.1, and Open-Sora 1.2.

**Compliance With Llm Reviewing Policy:**

Affirmed.

**Key Questions For Authors:**

1. The ablations are informative but still leave questions: How sensitive are results to τ scaling across layers/timesteps?
2. The process/measurement covariances (Q, R) appear time-invariant in the implementation, despite claims of a time-dependent noise schedule; this may limit adaptivity across diffusion timesteps known to have non-stationary dynamics ?

**Limitations:**

yes

**Strengths And Weaknesses:**

Strengths:
1. Recasting per-block feature-difference prediction as a state estimation problem is a fresh and principled perspective on runtime caching for VDMs.
2. The use of a recursive estimator with uncertainty (state + covariance) and an initial alignment phase offers a coherent alternative to both offline calibration and zero-order test-time heuristics.
3. Providing a simple, closed-form variance-minimizing update and steady-state analysis gives the approach some theoretical grounding beyond heuristics.

Weaknesses:
1. The process/measurement covariances (Q, R) appear time-invariant in the implementation, despite claims of a time-dependent noise schedule; this may limit adaptivity across diffusion timesteps known to have non-stationary dynamics.
2.Notation and timing indices (t, t+δ) occasionally cause confusion about how ΔI_t is computed at runtime in a skip/compute pipeline; the description would benefit from a more explicit per-block, per-timestep algorithm.
3. In terms of results, the performance gains do not substantially surpass the existing state-of-the-art.

---

> ### Author Rebuttal · Authors · 2026-03-29
>
> **We sincerely thank you for the encouraging assessment of our work and for providing such insightful, constructive comments. Per your suggestions, the detailed discussion below have been incorporated into the revised manuscript.**
>
> > W1 & Q2. Non-stationary dynamics
>
> We want to clarify that NaviCache demonstrates strong adaptability to non-stationary diffusion dynamics. The true adaptability of our system stems from:
>
> 1. **Handling Extreme Non-stationarity via Initial Alignment:** The most violent distribution shifts (high turbulence) occur in the very early denoising steps. Rather than engineering a complex, model-specific time-varying $Q_t$ to track this chaos, NaviCache handles it robustly through the **Initial Alignment** phase. By enforcing exact computation for the first $N_{align}$ steps, we bypass the most severe non-stationarity and initialize a mathematically stable unbiased state $\hat{x}_0$ and uncertainty covariance $P_0$.
>
> 2. **Dynamic Adaptivity via Posterior Covariance ($P_t$) and Calibrated Fusion Factor ($K_t$):** During the self-calibration stage, even with constant $Q$ and $R$, the system remains highly adaptive. The prediction uncertainty $P_{t|t-1}$ grows recursively during block-skipping phases due to the accumulation of the process noise $Q$. When the uncertainty threshold is breached, the **Calibrated Fusion Factor (CFF, $K_t$) automatically adjusts**. In relatively stable regimes, $K_t \rightarrow 0$ to maintain momentum; in turbulent regimes where prediction uncertainty is high, $K_t \rightarrow I$ to rapidly anchor to the new observation. This recursive uncertainty tracking mechanism provides the required adaptivity without needing explicit time-varying schedules.
>
> As demonstrated in our experiments, NaviCache proved remarkably robust based on different models (HunyuanVideo, Wan, Open-Sora).
>
> Furthermore, as shown in https://anonymous.4open.science/r/NaviCache_-C127/R3_csEP/w1/,
>
> 1. Diverse models/prompts exhibit a "flight trajectory" pattern: initial/final steps show large variations (takeoff/landing), while intermediate steps remain stable (cruising), confirming our navigation-inspired method's applicability (trajectory_{model}.png).
>
> 2. We have also evaluated the applicability of our method on samples with substantial 'trajectory' variations. (hard_sample_performance.png)
>
>
>
> > W2. Pseudo code.
>
> We have provided a detailed pseudo-code in the revised paper (https://anonymous.4open.science/r/NaviCache_-C127/R3_csEP/w2/).
>
>
>
> > W3. Performance comparison.
>
> To ensure a strictly fair comparison, we explicitly aligned the inference latency across all methods. Under similar time constraints, NaviCache consistently outperforms existing baselines, and its timestep skipping mechanism is significantly more adaptive, enabling it to skip more steps while still achieving superior performance (please refer to: https://anonymous.4open.science/r/NaviCache_-C127/R3_csEP/w3/). Specifically:
>
> - **On Wan 2.1-1.3B:** Compared to the most relevant SOTA offline calibration-free method (EasyCache), our NaviCache-fast achieves a significantly higher PSNR of 23.46 (+0.5 dB) at a faster latency of 96.40s, whereas EasyCache only reaches 22.96 at 99.21s.
> - **On HunyuanVideo:** Our NaviCache-mid achieves a higher PSNR of 32.65 at 1089.43s, outperforming EasyCache (which reaches a PSNR of 32.53 at 1100.30s) while requiring less inference time.
>
> Offline-calibrated methods (TeaCache) overfit to the priors of their calibration datasets, giving them a dataset-specific advantage that allows them to maintain a marginal lead in the VBench score. NaviCache is entirely zero-shot and runtime-adaptive; it trades this marginal overfitting benefit for true out-of-distribution generalization. It is worth noting that our method not only outperforms "Offline Calibration-free" methods. Even when compared to offline calibration-based methods like MagCache and TeaCache, our approach maintains an overall superiority (with the exception of TeaCache's VBench score—a metric where other baseline acceleration methods also fail to surpass TeaCache).
>
> **In summary, by eliminating the need for offline calibration, we avoid substantial time consumption as well as the strong assumption that the distributions of calibration and test data are closely aligned, all while achieving optimal performance under the same latency.**
>
>
>
> > Q1. About $\tau$
>
> Table 3 in our paper has already presented the relevant results. We have conducted a more detailed experimental analysis on $\tau$. (https://anonymous.4open.science/r/NaviCache_-C127/R3_csEP/q1/)

---

### Official Review · Reviewer_YrWj · 2026-03-13

**Soundness:** 4
**Presentation:** 3
**Significance:** 3
**Originality:** 3
**Overall Recommendation:** 5
**Confidence:** 3

**Summary:**

This paper proposes NaviCache, an offline-calibration-free method that reformulates feature evolution as an online recursive state estimation problem at test time. Specifically, the method treats the ratio between input change and output change as a latent state evolving along the diffusion trajectory, and realizes test-time self-calibration through mechanisms such as Initial Alignment, prediction-correction updates, and adaptive skip/update decisions. Experiments on HunyuanVideo, Wan2.1, and Open-Sora 1.2 show that NaviCache achieves a favorable speed-quality tradeoff compared with state-of-the-art baselines such as EasyCache, TeaCache, and MagCache.

**Compliance With Llm Reviewing Policy:**

Affirmed.

**Final Justification:**

The authors addressed my concerns in the rebuttal and helped me recognize the value of the work. Therefore, I have decided to give it an Accept rating.

**Key Questions For Authors:**

**Calibration-free?** Although the method is described as calibration-free, several important hyperparameters still appear to be manually chosen, including $N_{\text{align}}$ and the threshold $\tau$. Moreover, the threshold settings differ across backbones and across the fast/mid/slow modes.

1. Could the authors clarify why these values are selected in this way,
2. Whether there is any principled or automatic mechanism for adapting $\tau$ rather than tuning it separately for each model and mode?

**Relation to Kalman-style estimation.** The paper formulates test-time self-calibration using a recursive Bayesian filter, which is acceptable. However, the proposed framework appears highly consistent with classical Kalman-style recursive estimation, given the explicit state-space formulation, gain-based prediction/correction updates, and Riccati-style analysis.
In particular, the coefficient $K_t$, referred to as the *Calibrated Fusion Factor*, seems to play essentially the role of a Kalman gain. Could the authors discuss more explicitly how the method relates to standard Kalman filtering?

**Limited speedup on Open-Sora.** I am also curious why the speedup on Open-Sora appears less pronounced than on the other backbones. A brief explanation or analysis of this behavior would be helpful.

**Limitations:**

Yes.

**Strengths And Weaknesses:**

**Strengths**

1. The experimental design is well motivated and strengthens the paper. In particular, the method is evaluated on three different backbones (HunyuanVideo, Wan2.1, and Open-Sora), rather than being demonstrated on only a single model. This is a clear plus, since it makes the empirical evidence more convincing and reduces the concern that the method may only work well in one specific setting.

2. NaviCache introduces a meaningful conceptual shift from heuristic caching to online state estimation. The method itself forms a closed-loop system, including initialization, recursive prediction, observation-based update, and a skip/update gating mechanism, rather than stopping at a high-level intuition.

**Weaknesses**

1. Although the paper emphasizes that the method is offline calibration-free, it is not truly parameter-free or fully plug-and-play in a strict sense. The implementation still requires different choices of $N_{\text{align}}$ and $\tau$ for different backbones. In particular, while the paper uses a unified setting of $Q = R = 0.05I$, the threshold $\tau$ is still manually specified for each model and each operating mode, including the fast, mid, and slow settings.

2. Although the experiments are generally complete, the depth of diagnostic analysis is still limited. For example, the paper does not provide more detailed analyses of skip/update frequency, the additional overhead introduced by the state estimation module itself, or how performance varies under different motion complexity, video length, or resolution settings.

3. The writing and presentation are not fully polished in some places. One noticeable example is that the title is not fully consistent across pages: the first page uses *NaviCache: Test-Time Self-Calibration Caching for Video Generation*, while later page headers use *NaviCache: Trajectory-Aware Caching for Video Diffusion Models*, which suggests that the manuscript may not yet be completely unified.

4. While the empirical results are overall positive, the method does not show a clearly dominant advantage across all metrics and all backbones. For example, TeaCache still achieves slightly better VBench scores in some settings, and on Open-Sora 1.2 the gap between NaviCache-mid and EasyCache is relatively limited. This does not undermine the value of the paper, but it suggests that statements such as ``consistently outperforms baselines'' may be somewhat stronger than what the current evidence fully supports.

---

> ### Author Rebuttal · Authors · 2026-03-29
>
> **Thank you for your positive feedback and valuable suggestions. We have carefully addressed these points, and has added them into the paper.**
>
> > W1&Q1.Offline Calibration
>
> 1. Calibration-Free&$N_{align}$:
>
> "Offline calibration-free" means NaviCache eliminates the offline calibration process on datasets (unlike TeaCache and MagCache). But this doesn't mean parameter-free. $N_{align}$ integrates calibrator initialization from control theory with the trajectory characteristics of video generation (i.e., fluctuating & stabilizing, akin to airplanes' takeoff & cruise) (https://anonymous.4open.science/r/NaviCache_-C127/R2_YrWj/w1/). A brief alignment is required before autonomous state tracking begins to establish a mathematically stable prior. Another consideration is the fair comparison discussed below.We also analyzed it in Table2 of the paper
>
> 2. About $\tau$:
>
> A principled automatic mechanism—such as dynamically decaying $\tau_t$ based on the noise schedule (e.g., using a strict threshold during early structure generation and a relaxed one for later denoising)—is highly feasible
>
> But we intentionally excluded this automatic adaptation in our evaluation. An adaptive $\tau$ inherently makes the total skipped steps and inference latency input-dependent and uncontrollable.Trading significantly increased latency for marginal improvements is unfair to the baselines
>
> So we manually adjusted $\tau$ to align our inference time as closely as possible with baselines to ensure fair comparison. This yields a more convincing conclusion: under similar latency, NaviCache generally performs better
>
> In actual deployment where strict benchmarking is no longer a constraint, the aforementioned adaptive $\tau$ perfectly fits our theory: using a strict $\tau$ during the "takeoff" phase and a relaxed one during "cruise". Users can also configure this to achieve personalized speed-quality trade-off.
>
> > W2. Deeper analysis
>
> We added the detailed analysis at: https://anonymous.4open.science/r/NaviCache_-C127/R2_YrWj/w2/
> Frequency(1_skip_frequency.png, 1_skip_frequency_tab.png), Overhead(2_additional_overhead.png), Complexity(3_motion_complexity_hard_sample_performance.png), Length/Resolution(4_video_length_resolution_speedup.png, 4_video_length_resolution_performance.png)
>
> > W3.Inconsistency
>
> We aligned this with the title and ensured paper-wide consistency
>
> > W4&Q3.Statement&Open-Sora1.2
>
> We revised it to:"achieves a competitive speed-quality trade-off, generally demonstrating significantly superior performance under the similar latency"
>
> TeaCache overfits to calibration datasets, yielding dataset-specific advantages.Though offline calibration-free, NaviCache generally achieves significantly superior performance over calibration-dependent TeaCache and MagCache (other baselines similarly struggle to surpass TeaCache's VBench score)
>
> Under strictly aligned latency,NaviCache generally outperforms baselines: its adaptive skipping strategy skips more steps while performing better (https://anonymous.4open.science/r/NaviCache_-C127/R2_YrWj/w2/),especially on Wan2.1&HunyuanVideo
>
> Accelerating Open-Sora1.2 is relatively difficult.The model uses very few diffusion steps($T=30$),making it difficult to skip many timesteps.Further, its trajectory is relatively smooth,which makes it harder to leverage the advantage of NaviCache's real-time self-calibration(https://anonymous.4open.science/r/NaviCache_-C127/R2_YrWj/w1/)
>
> > Q2.About Kalman
>
> Our objective is to engineer a practical realization of Recursive Bayesian Estimation for video generation. While we drew inspiration from its classic form—the Kalman filter—it's entirely distinct from NaviCache's application in VDMs:
>
> **State Abstraction(Mapping):** Kalman filters linearly track absolute physical vectors(e.g., $X_t \in \mathbb{R}^6$ in 6-DoF kinematics).Conversely, tracking raw $d$-dimensional spatial VDM tensors($X_t \in \mathbb{R}^{c \times h \times w}$) is mathematically intractable due to extreme high-dimensional variance.NaviCache employs a severe non-linear abstraction, mapping state to a 1D scalar of $L_1$ norm ratios($x_t=\|\Delta O_t\|_1 / \|\Delta I_t\|_1$). So we filter the manifold's local change rate rather than spatial coordinates
>
> **Event-Triggered Measurement(Intermittent Observation):** Standard filters assume continuous measurements ($z_t=Hx_t+v_t$) and unconditionally execute Riccati corrections. NaviCache operates conditionally: the expensive measurement update occurs *only* if the accumulated error exceeds a safety boundary ($\mathcal{E}\_{acc} \ge \tau$). During skipped steps, the system relies purely on prediction ($\hat{x}\_{t|t} \equiv \hat{x}_{t|t-1}$), strictly accumulating process noise $Q$ without collapsing via $R$
>
> **Noise Semantics:** Unlike physical systems where noise matrices ($Q,R$) denote tangible sensor inaccuracies, our sensorless framework mathematically repurposes them to model intrinsic VDM latent feature turbulence and layer-wise computational variance

---

> > ### Author Rebuttal · Reviewer_YrWj · 2026-04-03
> >
> > The authors have fully addressed my concerns, and I will raise my score accordingly.

---

> > > ### Author Response · Authors · 2026-04-03
> > >
> > > Thank you so much for getting back to us so quickly. We are incredibly grateful for your positive feedback on our work, and we have fully integrated all the discussed content into the revised paper.

---

### Official Review · Reviewer_kjo7 · 2026-03-13

**Soundness:** 3
**Presentation:** 3
**Significance:** 3
**Originality:** 3
**Overall Recommendation:** 4
**Confidence:** 4

**Summary:**

This paper proposes NaviCache, a training-free caching framework for accelerating inference in video diffusion models. The method focuses on predicting whether the output of a Diffusion Transformer (DiT) block at a given timestep can be skipped by estimating the relationship between input differences and output differences during the diffusion process. Instead of relying on static mappings or offline calibration, the method models the evolution of the output-to-input change ratio as a trajectory and performs test-time self-calibration using a state estimation mechanism. Based on the estimated ratio and accumulated prediction error, the system dynamically decides whether to reuse cached outputs or perform full computation, aiming to improve inference efficiency while preserving visual quality.

**Compliance With Llm Reviewing Policy:**

Affirmed.

**Final Justification:**

My concern was addressed, and I recommend weak acceptance.

**Key Questions For Authors:**

See weaknesses

**Limitations:**

yes

**Strengths And Weaknesses:**

Strengths:

- The idea is simple and intuitive, and the overall framework is easy to follow. The pipeline mainly consists of an initialization stage, a state estimation module, and a skip decision mechanism.

- The method is training-free and plug-and-play, allowing it to be applied on top of existing video diffusion models without additional model training.

- The empirical results demonstrate improvements over previous caching approaches (e.g., TeaCache, MagCache, EasyCache) in terms of prediction accuracy for output differences and the overall quality-efficiency trade-off.

Weaknesses:

- The algorithmic novelty appears a bit limited. The method mainly replaces heuristic estimation of output differences with a state estimation formulation, which can be viewed as an engineering refinement rather than a fundamentally new approach.

- The key assumption that the ratio between output differences and input differences evolves smoothly along the diffusion trajectory is not rigorously justified. It would be helpful to provide more empirical analysis across different models, prompts, or schedulers to support this assumption.

- Several figures are difficult to read. E.g. the plots in Fig. 1 and Fig. 2 appear somewhat blurry or small. Improving the figure resolution and readability would significantly help.

---

> ### Author Rebuttal · Authors · 2026-03-29
>
> **We deeply appreciate your positive feedback on our work, as well as the insightful and constructive suggestions provided. The detailed discussion below has been added to the revised paper.**
>
> > W1. algorithmic novelty
>
> We respectfully clarify that our core novelty is introducing a fundamentally **new modeling paradigm** for test-time caching in video diffusion models. **As Reviewer YrWj noted,** 'NaviCache introduces a meaningful conceptual shift from heuristic caching to online state estimation. The method itself forms a closed-loop system, including initialization, recursive prediction, observation-based update, and a skip/update gating mechanism, rather than stopping at a high-level intuition.' **Similarly, Reviewer csEP highlighted that** 'Recasting per-block feature-difference prediction as a state estimation problem is a fresh and principled perspective on runtime caching for VDMs.
>
> As outlined in our core contributions and commended by other reviewers, our framework effectively bridges this gap through the following innovations, adaptively skipping timesteps to achieve superior performance in similar or even less latency:
>
> - **A Paradigm Shift in Modeling:** We abandon the instantaneous zero-order approximations and offline-fitted polynomials of prior works. Instead, we reformulate VDM feature evolution as a dynamic state-space model, establishing the first formal mapping between diffusion dynamics and Inertial Navigation System (INS) theory.
> - **Practical Adaptation for VDMs:** To make recursive Bayesian estimation viable for diffusion, we introduced a Novel State Mapping. Since raw spatial features in DiTs vary wildly, we mapped the state to the magnitude ratio of feature changes ($\Delta O_t / \Delta I_t$) to extract a stable, trackable trajectory. Furthermore, to handle Discontinuous Observations (because adaptively skipped blocks inherently lack exact observation data), we designed the unique Uncertainty-Aware Safety Gate logic to update the system systematically. **Finally, because** standard estimators instantly diverge when initialized in the high-turbulence zones of early diffusion steps, we introduced the Initial Alignment phase to mathematically calibrate the initial state and covariance matrix, completely stabilizing the non-stationary dynamics.
> - **Theoretical Error Guarantee:** Beyond empirical design, our formulation allows for rigorous mathematical analysis. We theoretically prove that our dual-state estimation engine achieves a strictly lower estimation error bound than the zero-order heuristics used by existing calibration-free methods (e.g., EasyCache).
>
>
>
> > W2. Assumption of smooth evolution
>
> We agree that visualizing trajectories under various conditions strengthens our premise. We have included trajectory visualization results across different conditions (https://anonymous.4open.science/r/NaviCache_-C127/R1_kjo7/w2/). In the figures, the solid lines represent the central trajectories, while the lines on either side represent the more marginal trajectories. We compared all prompts across different models (Hunyuan, Wan, OpenSora) and different schedulers (UniPC, DPM++). We found that under various conditions, all trajectories consistently align with the hypothesis that 'the ratio between output differences and input differences evolves smoothly along the diffusion trajectory.' The rate of feature change maintains distinct momentum and structural continuity, which demonstrates the broad applicability of our theory.
>
>
>
> > W3. Figure Readability.
>
> We have addressed this issue and uploaded the revised figure to https://anonymous.4open.science/r/NaviCache_-C127/R1_kjo7/w3/ . (Please note that the image preview in the anonymous repository may compress the visual quality; however, you can view the clear, high-resolution original figure by clicking "View raw" or "Download file".)

---

> > ### Author Rebuttal · Reviewer_kjo7 · 2026-04-03
> >
> > Thanks for the author's rebuttal. I'm more confidence to accepting the paper. please use higher-res figures for the revision.

---

> > > ### Author Response · Authors · 2026-04-03
> > >
> > > Thank you so much for getting back to us so quickly. We have ensured that all high-definition figures are added to the revised paper. Furthermore, the PNG images from the anonymous repository, originally intended for easy previewing, have also been embedded in the manuscript as PDFs.
> > >
> > > We were very encouraged to read your comment stating, “Fully resolved - My concerns have been adequately addressed,” along with your “more confidence to accepting the paper.” Given your positive evaluation, we wanted to send a friendly reminder just in case updating the score in the system slipped your mind~

---

### Decision · Program_Chairs · 2026-04-30

**Decision:**

Accept (regular)

**Comment:**

This paper proposes NaviCache, a training-free, plug-and-play caching framework that reformulates feature evolution in video diffusion models as an online recursive state estimation problem to enable adaptive block skipping at test time.

All three reviewers acknowledge the method as technically sound, well-motivated, and practically valuable, with consistent empirical improvements in speed–quality trade-offs across multiple modern video diffusion backbones (HunyuanVideo, Wan2.1, and Open-Sora) compared to prior caching approaches.

While concerns were raised regarding the level of algorithmic novelty, reliance on certain manually specified hyperparameters, and somewhat limited diagnostic analyses, reviewers agreed that the proposed framework represents a principled advancement over existing heuristic caching strategies.

The authors’ rebuttal satisfactorily addressed key technical questions regarding calibration, estimation design, and implementation details.

Overall, given its solid empirical validation, theoretical grounding, and potential to serve as a practical inference-time acceleration component for diffusion-based video generation systems, ACs recommend acceptance.